# Comparison of kinesiotape, counterforce brace, and corticosteroid injection in patients with tennis elbow: A prospective, randomized, controlled study

Seyyed Mohammadreza Hoseini[1], Mohammad Taghipour[2]*, Rahmatollah Jokar[3], Mehrdad Mostafaloo[4], Hoda Shirafkan[5], Khodabakhsh Javanshir[6]

1 Rehabilitation Research Center, Department of Physiotherapy, School of Rehabilition Sciences, Iran University of Medical Sciences, Tehran, Iran, 2 Mobility Impairment Research Center, Health Research Institute, Babol University of Medical Sciences, Babol, Iran, 3 Clinical Research Development Unit of Shahid Beheshti Hospital Department of Surgery, Babol University of Medical Sciences, Babol, Iran, 4 Clinical Research Development Unit of Shahid Beheshti Hospital, Department of Radiology, Babol University of Medical Sciences, Babol, Iran, 5 Department of Community Medicine, School of Medicine, Social Determinants of Health Research Center, Health Research Institute Babol University of Medical Sciences, Babol, Iran, 6 Department of Physiotherapy, School of Rehabilitation, Mobility Impairment Research Center, Health Research Institute, Babol University of Medical Sciences, Babol, Iran

* taghipourm@yahoo.com

## Abstract

### Objective

The aim of the study was to compare the effects of corticosteroid injection, kinesio tape, and counterforce brace on pain intensity, Common Extensor Tendon thickness, grip strength, and functional status in the treatment of lateral epicondylitis.

### Design

A total number of 51 patients were randomized into three groups. Group 1 was given kinesio tape, group 2 received a corticosteroid injection, and group 3 received a counterforce brace. Pain was measured using a visual analog scale, common extensor tendon thickness was measured with ultrasonography, functional status was measured using the disabilities of arm, shoulder, and hand questionnaire and grip strength was measured using a dynamometer. All evaluations were performed before treatment and at the second and fourth weeks after the treatment.

### Results

No significant differences between the groups were observed for the visual analog scale scores, common extensor tendon thicknesses, grip strength, and disabilities of arm, shoulder, and hand questionnaire score compared to the baseline (P > 0.05). A statistically significant difference was found between the pretreatment and post-treatment evaluations of pain intensity and common extensor tendon thickness in all

**Data availability statement:** All relevant data are within the manuscript.

**Funding:** The author(s) received no specific funding for this work.

**Competing interests:** The authors have declared that no competing interests exist.

groups at the second and fourth weeks after treatment. According to the disabilities of arm, shoulder, and hand questionnaire scale, the condition improved significantly in the brace group and corticosteroid group, whereas it had not improved in the kinesiotaping group compared to before treatment. None of the treatment methods increased the patients' grip strength (P > 0.05).

## Conclusion

The corticosteroid injection, kinesiotape, and specially counterforce brace effectively reduced pain and tendon thickness. However, none of these treatment methods were superior to the others.

## Trial registration

This study was registered in the Iranian Registry of Clinical Trials (IRCT20091214002851N7).

## Introduction

Lateral epicondylitis (LE), known as tennis elbow [1], is the primary etiology of elbow pain [2]. The dominant arm is the most commonly affected, with a prevalence of 1–3 percent in the general population [3]. The pain is usually localized in the epicondyle [4], and is usually triggered by exerting pressure on the epicondyle, stretching the extensor muscles, and resisting wrist and/or third finger extension. In severe cases, the pain can extend to the shoulder and wrist [5]. Currently, we believe that tennis elbow is not inflammatory, with neovascularization, collagen degeneration, fibroblast proliferation, and mucoid degeneration representing the pathologies [6]. An overload injury or microtrauma causes the disease of the common extensor tendon (CET) near its attachment to the lateral epicondyle [7]. Ultrasonography (US) is an important instrument in the fields of sports medicine and rheumatology, and it is frequently used as a standard measurement in clinical trials [8]. US is a commonly used and affordable imaging method that allows for the evaluation of tendon abnormalities without the need for invasive procedures [9].

Several interventions are commonly employed to manage tennis elbow, including kinesio tape (KT), counterforce brace, and corticosteroid injection (CSI). CSI is a commonly used approach for treating LE [10,11]. The mechanism of action underlying CSI treatment remains uncertain [12]. Previous studies have indicated that CSI is beneficial in the short term, but they are detrimental in the long term and have a higher chance of recurrence [13,14]. Following the injection, side effects may include osteomyelitis, cellulitis, ecchymosis, subcutaneous adipose tissue atrophy, and hypopigmentation [15].

Conservative management is recommended as the initial treatment for LE and is considered to be successful in the majority of patients [16]. One of these treatments is KT. KT is a common management technique for a range of musculoskeletal disorders [17]. Researchers have proposed several potential effects of KT, including pain

alleviation, restoration of normal muscle function, enhancement of proprioceptive feedback, and correction of articular malalignment [18,19].

The use of a proximal forearm brace also referred to as a "counterforce brace," has been one of the more popular treatments [20]. Despite the popularity of counterforce brace, its mechanism of action is still debated. The theory behind many tennis elbow treatments is to unload the origin of the CET, allowing it to recover and heal [21]. Bracing can accomplish this by applying pressure to the extensor bundle, reducing and spreading some of the stress away from the extensor origin [22]. Alternatively, the brace may restrict the complete expansion of the extensor muscles while they contract, reducing the maximum force that may be generated at the extensor origin [23]. While these interventions are commonly applied in clinical practice, the evidence supporting their individual effectiveness remains inconsistent, and no prior randomized study has directly compared all three within a single controlled trial. Moreover, the present study incorporated ultrasonographic evaluation of tendon thickness as an objective measure, providing structural data beyond subjective assessments of pain and functional status.

Therefore, the aim of this study was to compare the short-term effects of KT, counterforce brace, and CSI on pain, tendon thickness, grip strength, and functional status in patients with lateral epicondylitis.

## Methods

The study was a prospective, randomized, assessor-blinded clinical trial. We prospectively followed up all patients after they gave written informed consent prior to data collection. Data were collected at the Orthopedic clinic of Babol University of Medical Sciences, Babol, Iran. The study participants were recruited from July 11, 2022, to April 11, 2023, with follow-up continuing until May 11, 2023. Additionally, the authors considered all the recommendations disclosed in the Declaration of Helsinki to ensure the rights of the participants. The study design was approved by the local Clinical Ethics Committee of Babol University of Medical Sciences (IR.MUBABOL.HRI.REC.1401.097) and registered in the Clinical Trial Registry (IRCT20091214002851N7). No important changes were made to the methods after trial commencement.

The sample size was calculated based on the study by Oken et al. [21], using pain intensity (VAS) as the primary outcome. Assuming a between-group effect size of 0.15, we planned a repeated measures ANOVA for analysis. To account for an anticipated dropout rate of up to 10%, we enrolled 51 patients (33 females and 18 males) aged 29–55 years (mean±SD: 43±7) to ensure adequate power. With 17 patients per group, the study would have 80% power and a 95% confidence interval. No interim analyses or stopping rules were applied in this study.

### Inclusion criteria

Patients had to fulfill the following requirements to be eligible for the clinical trial: be at least eighteen years old, have a clinical diagnosis of LE verified by positive results from specific testing, such as the Maudsley, Mill, or Cozen tests, and have had symptoms for four weeks to six months before presentation [24,25].

### Exclusion criteria

Patients who met any of the following criteria were excluded from the study: cervical radiculopathy, neuropathy, previous diagnosis of rheumatological disease, diabetic neuropathy, history of elbow surgery, pregnancy, allergic reactions to tape, elbow joint arthrosis, or previous corticosteroid injections in the elbow region [26,27].

### Treatment procedure

This study followed the CONSORT guidelines for reporting randomized controlled trials. Randomization was performed using the permuted block randomization method with a block size of six to ensure balanced allocation across groups. Each block contained an equal number of participants for each treatment group, and the order of the three treatments within each block was randomized by a statistician. To ensure allocation concealment, sealed envelopes labeled with

the letters A, B, and C were prepared. Patients randomly selected an envelope, which determined their treatment group. Patients who picked an envelope labeled "A" were assigned to the counterforce brace group, those who picked "B" were assigned to the KT group, and those who picked "C" were assigned to the CSI group. Each envelope had a randomly generated code, which was recorded in the patient's file (checklist) upon enrollment (Fig 1).

Within the KT group, we implemented a KT treatment for LE using two Y-shaped Kinesio strips, following the method suggested by Kaze et al [28]. The main strip is placed along the extensor muscles, while the secondary strip is positioned vertically to the first strip over the upper part of the forearm. The main strip was employed to inhibit the targeted muscles, while the secondary strip corrected the fascia. The tape was applied to the distal insertion of the muscle and continued to its origin at the elbow level on both legs. The tape was anchored using five-centimeter distal and proximal sections without tension. Following this, the Y-shaped tape was fully tensioned over the muscle origins at the lateral epicondyle, except the 5-cm anchor sections. The tape applications were performed by a registered and trained physiotherapist and were

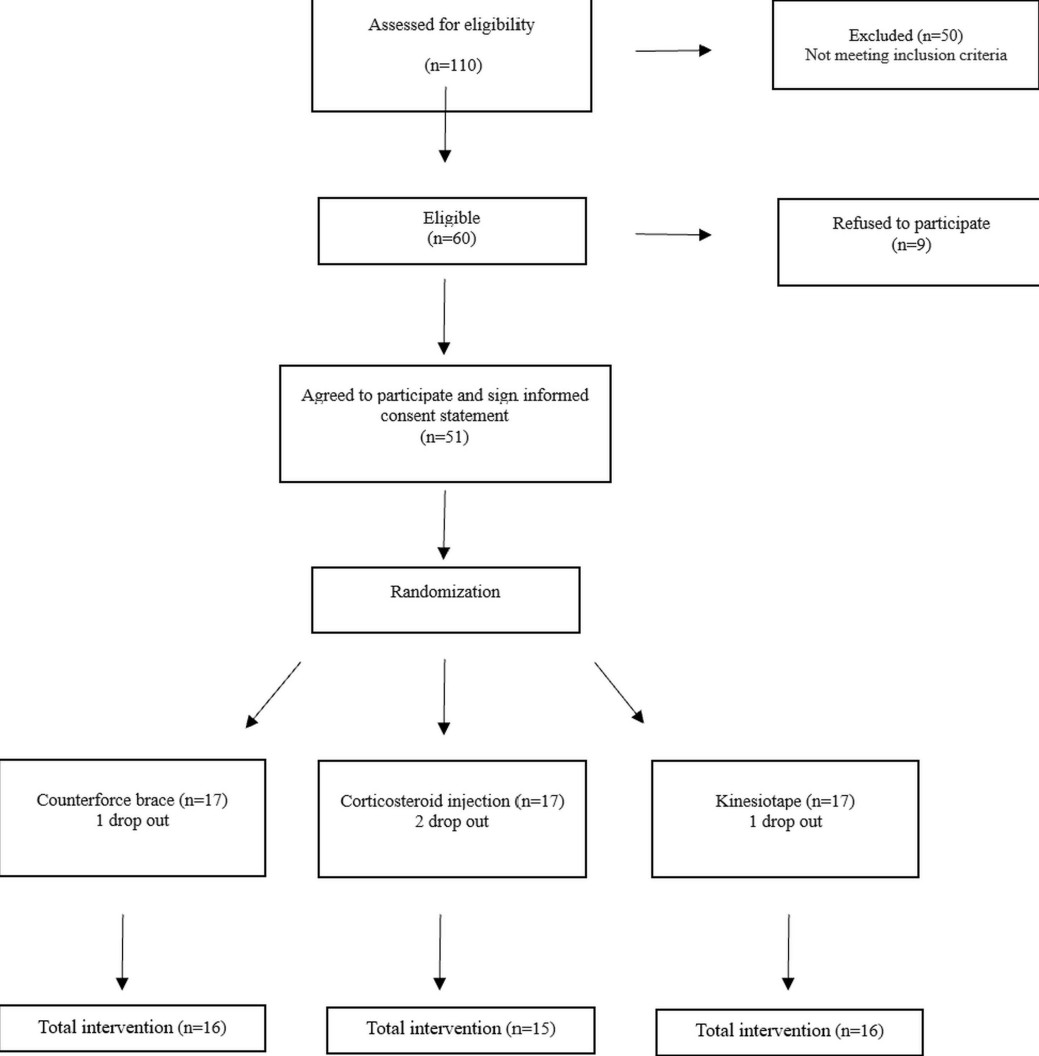

**Fig 1. Flowchart diagram for the participants who were randomized into three groups.**

changed every three days for two weeks, resulting in a total of four applications for each patient [29]. Before each subsequent application, the skin was carefully cleaned and dried, and the patients were instructed to restrict their contact with water [18] (Fig 2).

In the second group, patients received a CSI consisting of 1 ml of triamcinolone (20 mg/ml) with 1 ml of 1% lidocaine into the painful lateral epicondyle. The needle was entered in a vertical direction until it pierced the tendinous insertion and reached the bone. The entire lesion was infiltrated by partially withdrawing the needle and then re-inserting it. The injection was given to patients while their forearm was placed on a stable surface in a pronated position with a 45-degree elbow flexion. A orthopedist with approximately 15 years of experience performing musculoskeletal injections administered the injection. Each patient received a single injection [24].

In the third group of the study, the patient received a counterforce brace. Patients using the counterforce brace were told to find the most painful spot on their elbow and then apply the brace to the muscle belly, moving toward the wrist until it was 5 cm below the elbow crease. Patients were asked to flex and extend their elbows and clench their fists while wearing the brace to ensure it did not restrict their movement or grip strength. Patients were instructed to wear the brace continuously for 2 weeks, except for sleeping and showering [30].

We recommended that patients refrain from engaging in activities that have the potential to exacerbate their pain, such as lifting objects of substantial weight, performing household chores like laundry, or participating in activities that involve gripping or squeezing activities, such as kneading dough, utilizing small hand tools like pliers or screwdrivers, or engaging in gardening [31]. To ensure adherence, patients received detailed verbal instructions at baseline. The treating clinician conducted regular telephone follow-ups to monitor compliance and address any concerns. Additionally, protocol compliance was discussed at each follow-up visit. No specific exercises were prescribed for any patient in any of the treatment groups [32].

### Blinding and measurement parameters

The assessments were conducted by a masked assessor, who was a professional research assistant. Participants were instructed to refrain from disclosing any information about the treatment to the assessor. Although patients were not blinded, to minimize placebo effects and bias, patients were provided with uniform information regarding the effectiveness of each intervention, emphasizing that all interventions are widely accepted treatments for lateral epicondylitis. The patients were evaluated at baseline, 2 weeks later, and 4 weeks after their first intervention [18]. At baseline, demographic data was collected, such as age, sex, height, weight, employment status, and involvement in sports and leisure activities.

### Primary outcome

**VAS.** A visual analog scale was used to evaluate the intensity of pain reported by the patients. A 10 cm-long horizontal line was utilized as the scale; 0 cm on the left indicated no pain, and 10 cm on the right indicated the highest level of pain. We asked patients to mark the line at the point that represented their level of pain [33].

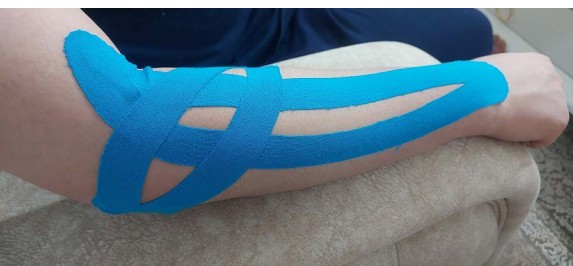

**Fig 2. Kinesio taping of lateral epicondylitis.**

### Secondary outcome measures

**Tendon thickness.** During the study, ultrasonography scanning was performed using a color Doppler Voluson E6 ultrasound system manufactured in the USA. Ultrasonography scanning was utilized to evaluate the thickness and echogenicity of the CET and the bony cortex of the lateral epicondyle (Fig 3). The imaging was conducted with patients in a seated position, with their elbows flexed at a 90-degrees, their wrists pronated, and their arms resting on the table [2]. The patients' ultrasonographic evaluation was conducted by a physician with ten years of experience in musculoskeletal ultrasonography. This clinician was blinded to the patients' clinical data, meaning that they were unaware of which patients had received specific treatments.

**DASH questionnaire.** The 30-item DASH questionnaire is self-reported and evaluates upper-limb symptoms and functional ability [24].

**Grip strength.** To assess maximal grip strength without pain [32], the final clinical outcome measure in the study involved using a handgrip dynamometer (Saehan Hydraulic Hand Dynamometer SH5001; Saehan, Changwon, Republic of Korea). During the test, participants were positioned in a chair with their shoulders flexed at a 90-degree angle, their elbows extended, and their forearms in a neutral position. The patients squeezed the dynamometer maximally for 3s. The average of three successive observations made 60 seconds apart, was recorded by the researchers [24]. No changes were made to the trial outcomes after the trial commenced.

### Statistical analysis

The SPSS 25 software (SPSS Inc., Chicago, 103 IL, USA) was used for data analysis. For data distribution, the Kolmogorov–Smirnov test was used. The Statistical analysis was performed using repeated measures ANOVA for the parameters with normal distribution, The post hoc Bonferroni analysis was applied to the parameters that were significant with the ANOVA test, and a p-value less than 0.05 was considered significant. For the outcome measurement obtained from VAS for pain, tendon thickness, grip strength and DASH, the repeated measures ANOVA, which has tests of between-subjects effects and tests of within-subjects effects (evaluation time: pretreatment, 2 wks after treatment, and 4 wks after treatment), were performed and analyzed by the intention-to-treat principle. No additional analyses, such as subgroup, were performed.

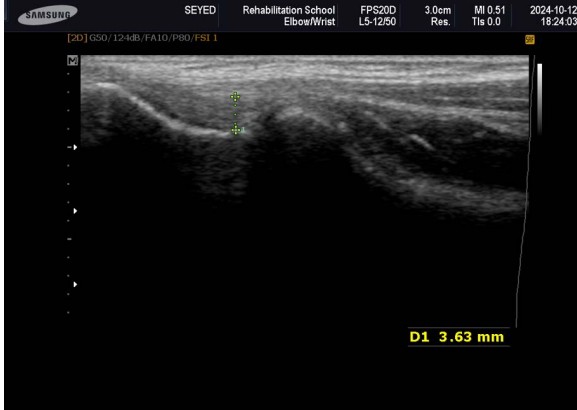

**Fig 3. Measurement of tendon thickness.**

## Result

Recruitment took place from July 11, 2022, to April 11, 2023, with follow-up assessments conducted until May 11, 2023. Except for four patients (one in the KT (unrelated shoulder injury), one in the counterforce brace group (inability to attend the clinic), and two in the CSI group (withdrew subsequently with no reason given)), all patients completed the study, totaling 47 patients (16 men, 31 women; mean age 52.43 years; range 29–55 years). The demographic and clinical characteristics of the three groups are presented in Table 1.

Gender, mean age, baseline VAS, tendon thickness, grip strength, and DASH questionnaire scale of the affected hand did not differ between the groups. The trial ended as planned after the completion of the 4-week follow-up period for all participants. The elbow pain assessed by VAS had improved significantly in all groups at the end of treatment, and this improvement continued in all groups up to the fourth week ($p < 0.05$) (Table 2). The CET thicknesses significantly decreased after 2 and 4 weeks in all groups ($P < 0.05$) (Table 3).

The grip strength of the affected hand had not increased in any group between two and four weeks after treatment (Table 4). According to the DASH questionnaire scale, the condition improved significantly in the brace group and CSI group, whereas it had not improved in the KT group compared to before treatment (Table 5). No significant differences between the groups were observed for the VAS scores, extensor tendon thicknesses, grip strength, and DASH questionnaire scale compared to baseline ($P > 0.05$). No ancillary analyses, such as subgroup or adjusted analyses, were conducted.

**Table 1. Characteristics of the Patients and Group comparing pretreatment.**

| Variable | Brace Group (n=17) | Kinesio tape Group (n=17) | Injection Group (n=17) | p |
|---|---|---|---|---|
| Age (yr) | 43.58±6.14 | 45±6.43 | 42±9.36 | 0.507 |
| Gender (women/men) | 12/5 | 11/6 | 9/8 | 0.556 |
| BMI(kg/m2) | 27.51±2.96 | 25.65±3.84 | 26.43±3.02 | 0.281 |
| Visual pain scale | 5.39±1.53 | 4.31±1.48 | 5.14±1.73 | 0.127 |
| Tendon thickness(mm) | 4.26±0.82 | 3.89±0.70 | 4.09±0.72 | 0.365 |
| Grip strength(kg) | 25.12±8.59 | 22.09±10.21 | 23.66±14 | 0.733 |
| DASH questionnaire | 67.30±18.36 | 63.52±22.87 | 64.65±16.96 | 0.847 |

**Table 2. Comparison of VAS between baseline and posttreatment values of the three treatment groups.**

| Visual pain scale Groups | Baseline Mean±SD | Week 2 Mean±SD | P | Week 4 Mean±SD | P | Effect Size | P* |
|---|---|---|---|---|---|---|---|
| Counterforce brace (n=17) | 5.39±1.53 | 3.71±2.07 | ≤0.001a | 2.92±2.34 | 0.001b | 0.579 | ≤0.001 |
| Kinesio tape(n=17) | 4.31±1.48 | 3.15±1.52 | 0.04a | 2.98±1.39 | 0.04b | 0.253 | 0.010 |
| Cortico steroid(n=17) | 5.14±1.73 | 3.30±2.17 | 0.008a | 2.71±1.92 | 0.002b | 0.447 | ≤0.001 |
| P-value** | 0.127 | 0.689 | | 0.911 | | | |

a=Comparison between pretreatment and 2-week posttreatment (within-group).

b=Comparison between pretreatment and 4-week posttreatment (within-group).

P*=Changes within each group over time.

P-value**= Between-group comparisons at each time point (baseline, week 2, and week 4).

The baseline P-value** was used to confirm group equivalence.

**Table 3. Comparison of tendon thickness between baseline and posttreatment values of the three treatment groups.**

| Tendon thickness(mm) | Baseline | Week 2 | | Week 4 | | Effect Size | P* |
|---|---|---|---|---|---|---|---|
| Groups | Mean±SD | Mean±SD | P | Mean±SD | P | | |
| Counterforce brace (n = 17) | 4.26±0.82 | 3.95±0.65 | 0.041a | 3.79±0.73 | 0.001b | 0.327 | 0.003 |
| Kinesio tape(n = 17) | 3.89±0.70 | 3.26±0.92 | 0.040a | 3.41±0.58 | 0.001b | 0.254 | 0.026 |
| Cortico steroid (n = 17) | 4.09±0.72 | 3.66±0.70 | 0.024a | 3.53±0.46 | 0.005b | 0.397 | 0.001 |
| P-value** | 0.365 | 0.039 | | 0.172 | | | |

**Table 4. Comparison of grip strength between baseline and posttreatment values of the three treatment groups.**

| Grip strength(kg) | Baseline | Week 2 | | Week 4 | | Effect Size | P* |
|---|---|---|---|---|---|---|---|
| Groups | Mean±SD | Mean±SD | P | Mean±SD | P | | |
| Counterforce brace (n = 17) | 25.12±8.59 | 25.58±8 | 1.000a | 27.30±8.18 | 0.430b | 0.089 | 0.226 |
| Kinesio tape(n = 17) | 22.09±10.21 | 22.33±10.33 | 1.000a | 22.26±9.79 | 1.000b | 0.029 | 0.584 |
| Corticosteroid(n = 17) | 23.66±14 | 24.23±11.90 | 1.000a | 22.91±10.18 | 0.211b | 0.012 | 0.688 |
| P-value** | 0.733 | 0.306 | | 0.248 | | | |

**Table 5. Comparison of the DASH questionnaire between baseline and posttreatment values of the three treatment groups.**

| DASH questionnaire | Baseline | Week 2 | Week 4 | Effect Size | P* | DASH questionnaire | Baseline |
|---|---|---|---|---|---|---|---|
| Groups | Mean±SD | Mean±SD | P | Mean±SD | P | | |
| Counterforce brace (n = 17) | 67.30±18.36 | 54.96±20.03 | 0.002a | 53.43±23.75 | 0.002b | 0.175 | 0.001 |
| Kinesio tape(n = 17) | 63.52±22.87 | 56.48±18.06 | 0.082a | 53.82±17.25 | 0.125b | 0.254 | 0.062 |
| Cortico steroid(n = 17) | 64.65±16.96 | 56.99±13.34 | 0.044a | 53.05±14.69 | 0.003b | 0.324 | 0.005 |
| P-value** | 0.847 | 0.939 | | 0.993 | | | |

## Discussion

The results of the present randomized clinical trial study showed that no significant difference between groups was detected for all outcome measures. All treatment interventions significantly reduced pain intensity and CET thickness at 2 and 4 weeks following the completion of the treatment. The functional status of the upper limb of the patients determined by the DASH questionnaire significantly improved at the end of the treatment in the Counterforce brace and CSI groups and after 4 weeks following the completion of the treatment. Another finding from our study was that none of the treatment methods increased the patients' grip strength.

Corticosteroids have the ability to reduce proinflammatory mediators and affect the cells involved in inflammatory responses, resulting in both anti-inflammatory and direct pain-relieving actions [34]. Within 24–48 hours, corticosteroids begin to exert their effects, and their duration of action is generally 2–3 weeks [35]. Coombes et al. suggested that corticosteroids may have an analgesic effect on neuropeptides, specifically substance P and calcitonin gene-related peptides, which are increased in tendinopathy [36]. Despite local therapy's efficacy, corticosteroids' side effects and problems have been more apparent in recent years [6]. According to a review article by Foye et al., around 15% of patients experienced adverse effects when receiving corticosteroid injections for tennis elbow. Among them, postinjection pain was most common, followed by skin atrophy and skin depigmentation [35]. Our study found no other adverse effects except for postinjection pain.

Recently, KT has gained popularity as a treatment for LE. KT increases the activation of motor units, induces proprioceptive feedback, and stimulates cutaneous mechanoreceptors [37]. Dilek et al. reported that there was an improvement in pain reduction, patient satisfaction, and disease staging at the end of the second and sixth weeks of their studies with 31 LE [38]. There have been variable results about the effect of KT on grip strength in patients with LE. According to Cho et al. KT improved functionality, increased grip strength, and decreased pain intensity after taping [27]. However, Giray et al. compared KT combined with exercises, sham taping, and exercises alone. In their study, Four weeks after treatment, the combination of tape and exercises proved more effective than other methods in reducing upper limb pain and disability [29]. The results of this study support our findings that there was no improvement in grip strength.

Patients in the KT group might have done less daily activity due to the constant feeling of tape on their elbows. Patients in the KT group did not have any taping sessions between the second and fourth weeks; However, an improvement was observed in pain intensity and CET.

Splinting is one of the conservative treatment modalities that is most commonly used to treat lateral epicondylitis. The forearm extensors' tension is decreased by lateral counterforce braces [39]. However, studies on the effectiveness of splinting in LE have also shown conflicting results. A systematic review using the Cochrane database revealed that there is insufficient evidence to draw definitive findings about the efficacy of braces in treating LE [40]. In a meta-analysis, Borkholder et al. reported 11 studies that provide early positive, but not conclusive, evidence supporting the effectiveness of bracing lateral epicondylitis [41]. In several studies, elbow straps or sleeve orthoses relieved pain and improved grip strength better than placebo orthoses or wrist splints [4, 42].

The same kind of brace that we used was compared by Struijs et al. with physical therapy or combination therapy (brace plus physical therapy). They found that only brace treatment was superior for problems during daily activities. According to their study conclusion, a brace can be regarded as an initial and supportive treatment [43]. Our results are in line with the short-term benefits of forearm bracing (disability, satisfaction, and severity of complaints). Holdworth and Anderson found that hydrocortisone phonophoresis and brace treatment significantly reduced resting pain in patients with LE when compared to conventional US [44]. According to Abdelmonem et al. in patients with tennis elbow pathology, the counterforce brace outperformed KT in terms of pain reduction and an increase in the myoelectric activity of the wrist flexors and extensors [45]. Furthermore, the use of the counterforce brace improved overall elbow functional activities and pain reduction (both frequency and intensity) in patients with tennis elbows compared to the placebo group in the study [30]. In contrast, a study by Phadke and Desai found that KT and counterforce brace had an equal impact on managing LE by decreasing pain and reducing disability [46].

The muscle tendons located in the lateral epicondyle have good blood circulation. So when too much stress is applied to this area, the body starts to create more tissue than it needs. For this reason, the tendon undergoes hypertrophy and increases in thickness [47, 48]. It is well recognized that ultrasonographic measurement is a crucial diagnostic tool that aids in the diagnosis of LE in conjunction with the clinical examination findings [1]. According to Gündüz et al., during the 6-month follow-up, CET thicknesses in the local steroid injection, extracorporeal shockwave therapy, and traditional physiotherapy groups did not change significantly from the baseline measurements in their study. They suggested that to obtain a significant result in CET thickness, further measurements were required following the 6-month follow-up [2]. Short-term reductions in tendon thickness may reflect decreased local edema rather than structural healing. Long-term studies are required to determine their prognostic value. In clinical practice, ultrasound is being used more often to diagnose tennis elbow. However, several factors can affect the accuracy of ultrasound in diagnosing tennis elbow, including the equipment, operator experience, and pathology stage [49].

To evaluate the isolated effects of each intervention, no concurrent exercise or rehabilitation program was included. Although this may reduce generalizability to typical clinical practice, it reflects real-world situations where some patients are unable or unwilling to engage in exercise. The lack of improvement in grip strength, despite reduced pain and tendon thickness, suggests that targeted strengthening is necessary for functional recovery.

Numerous studies that compare the safety and efficacy of various approaches to treating LE can be found in the literature.

However, the perfect treatment option has not been identified yet. LE can show a self-limiting attitude in some cases, whereas it can be refractory in others. The patients' occupations and activity levels may affect this outcome. This research provides a different approach to treatment. In recent years, as CSI have slowly lost their popularity [50], the safe use of Counterforce brace and KT has increased [19, 43].

The strengths of our study are the prospective and randomized design and the measurement of tendon thickness. However, as a limitation of the study, we lost 7.8% of the patients during the follow-up. Although some between-group differences did not reach statistical significance, the study's randomized controlled design, use of repeated measures ANOVA, and balanced baseline characteristics strengthen the reliability of the findings. The absence of statistical differences may reflect comparable clinical effectiveness among the interventions. However, limited statistical power and short follow-up duration may have also played a role. These findings should be interpreted with caution but still offer relevant guidance for clinical practice.

Given the short follow-up, the potential for recurrence, particularly after corticosteroid injection, remains unclear. Longer-term trials are needed to evaluate sustained outcomes.

The findings of this study may be generalizable to patients with lateral epicondylitis of similar age and symptom duration treated in outpatient settings, though applicability to other populations or settings may vary depending on patient characteristics and treatment availability.

Given the comparable outcomes, clinicians may base treatment choice on factors such as cost, availability, and patient preference, supporting a personalized approach to care.

## Conclusions

The results of the present study showed that Counterforce brace, kinesiotape, and corticosteroid injection were all effective in reducing pain and tendon thickness in patients with LE, and none of these treatment methods were superior to each other.

## Supporting Information

**S1 File. Trial study protocol.**
(PDF)

**S2 Data. Raw data file.**
(CSV)

**S3 Checklist. CONSORT checklist.**
(DOCX)

## Acknowledgments

We sincerely thank Dr. Tahere Seyedhoseinpoor for her generous support and collaboration throughout the research process. We also express our gratitude to Sekineh Kamali Ahangar, an expert at the Clinical Research Development Unit of Shahid Beheshti Hospital in Babol, for her valuable cooperation. The full trial protocol can be accessed through the Iranian Registry of Clinical Trials (IRCT) (https://irct.behdasht.gov.ir) using the registration number IRCT20091214002851N7.

## Author contributions

**Conceptualization:** Mohammad Taghipour.

**Data curation:** Seyyed mohammadreza Hoseini, rahmatollah jokar, Mehrdad Mostafaloo.

**Formal analysis:** Seyyed mohammadreza Hoseini, Hoda shirafkan.

**Investigation:** Seyyed mohammadreza Hoseini, rahmatollah jokar.

**Methodology:** Seyyed mohammadreza Hoseini, Mohammad Taghipour, rahmatollah jokar, Mehrdad Mostafaloo, Hoda shirafkan.

**Project administration:** Seyyed mohammadreza Hoseini, Mohammad Taghipour.

**Resources:** Seyyed mohammadreza Hoseini.

**Supervision:** Seyyed mohammadreza Hoseini, Mohammad Taghipour.

**Validation:** Seyyed mohammadreza Hoseini, Khodabakhsh Javanshir.

**Visualization:** Seyyed mohammadreza Hoseini, Mohammad Taghipour, Khodabakhsh Javanshir.

**Writing – original draft:** Seyyed mohammadreza Hoseini, Mohammad Taghipour.

**Writing – review & editing:** Seyyed mohammadreza Hoseini, Mohammad Taghipour.

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
