## [Decision Letter · Decision Letter 0]

Dear Dr. Taghipour,

Thank you for submitting your manuscript to PLOS ONE. After careful consideration, we feel that it has merit but does not fully meet PLOS ONE’s publication criteria as it currently stands. Therefore, we invite you to submit a revised version of the manuscript that addresses the points raised during the review process.

Kindly make the following changes:

Make sure the introduction articulates with the research question.Justify the rationale for selecting the reason for selecting these modalities, particularly when the literature shows uncertainty about their mechanism of action and effectiveness. Also there should be a strong need for the study to be conducted as there is adequate literature demonstrating clinical significance and novelty of the study.Please make sure the potential confounding variables are addressed in the manuscript.Please justify the sample size calculation. In terms of effect size assumed, which model was used and was patient drop out considered during sample size calculation.  As there was no exercise protocol included in the intervention period, please justify how its impact on the findings and its application in the real world.Please provide detailed explanation on how the authors made sure patient adherence and protocol compliance across three groups, particularly in KT and brace usage group.Please explain the measures taken to mitigate potential placebo effects.As the blinding was not done, please explain how patient bias was limited.Please expand on the significance of the findings, especially about the grip strength.As the follow-up was short term, please explain the long-term implications of the treatment on patient outcomes.Please explain how lack of exercise components may have influenced the grip strength.Please explain the clinical implication of the results, as all the three interventions had similar effects.Please explain if the tendon thickness changes have long term implications on functional recovery or it is a transient change.Please provide a detailed explanation on the statistically non-significant results.

Please make the other changes suggested by the reviewers.

We look forward to receiving your revised manuscript.

Kind regards,

Prateek Srivastav

Academic Editor

PLOS ONE

2. We note that you have selected “Clinical Trial” as your article type. PLOS ONE requires that all clinical trials are registered in an appropriate registry (the WHO list of approved registries is at https://www.who.int/clinical-trials-registry-platform/network/primary-registries " https://www.who.int/clinical-trials-registry-platform/network/primary-registries and more information on trial registration is at http://www.icmje.org/about-icmje/faqs/clinical-trials-registration/ ). Please state the name of the registry and the registration number (e.g. ISRCTN or ClinicalTrials.gov ) in the submission data and on the title page of your manuscript. a) Please provide the complete date range for participant recruitment and follow-up in the methods section of your manuscript. b) If you have not yet registered your trial in an appropriate registry, we now require you to do so and will need confirmation of the trial registry number before we can pass your paper to the next stage of review. Please include in the Methods section of your paper your reasons for not registering this study before enrolment of participants started. Please confirm that all related trials are registered by stating: “The authors confirm that all ongoing and related trials for this drug/intervention are registered”. Please see http://journals.plos.org/plosone/s/submission-guidelines#loc-clinical-trials for our policies on clinical trials.

3. We note that your Data Availability Statement is currently as follows: [All relevant data are within the manuscript]

Reviewers' comments:

Reviewer's Responses to Questions

**Comments to the Author**

1. Is the manuscript technically sound, and do the data support the conclusions?

Reviewer #1: Yes

Reviewer #2: Partly

2. Has the statistical analysis been performed appropriately and rigorously?

Reviewer #1: I Don't Know

Reviewer #2: No

3. Have the authors made all data underlying the findings in their manuscript fully available?

Reviewer #1: Yes

Reviewer #2: Yes

4. Is the manuscript presented in an intelligible fashion and written in standard English?

Reviewer #1: Yes

Reviewer #2: Yes

Reviewer #1: In addition to providing a response, please highlight and indicate the changes in the revised text of the article.

Dear Author, please recheck and answer the below items:

In the Introduction part:

Does the introduction clearly articulate a specific hypothesis or research question that guides the comparative analysis of the three interventions (KT, counterforce brace, and CSI)?

Have the authors sufficiently justified the rationale for selecting these three particular treatment modalities, especially given the noted uncertainty in their mechanisms and long-term effectiveness?

While prior literature is cited regarding the benefits and limitations of each treatment, do the authors adequately address potential confounding variables or population differences that may have influenced past findings?

Given the mention that no prior study has compared all three interventions simultaneously, could the authors elaborate more on the clinical significance and novelty of this comparison to reinforce the study’s contribution to the field?

In the Methods part:

Is the justification for the sample size calculation sufficiently clear, particularly in relation to the clinical relevance of the selected effect size (0.15) and the assumptions drawn from the Oken et al. study?

Given that no specific exercises or rehabilitation protocols were included during the intervention period, could this omission have impacted the comparability or external validity of the findings, especially for real-world application?

Can the authors provide more detail on how treatment adherence and protocol compliance were monitored across the three groups, particularly in interventions like KT and brace usage that depend heavily on patient behavior?

Were there any measures taken to assess or mitigate potential placebo effects, especially considering the visible and tactile nature of KT and bracing compared to the single CSI administration?

Since the blinding was limited to the assessor, how did the authors account for the potential influence of patient expectations or treatment preferences on subjective outcomes like VAS and DASH scores?

In the Discussion part:

Could the authors expand on the clinical significance of the findings, particularly considering that all interventions improved outcomes similarly but grip strength did not increase in any group? How might this inform treatment recommendations?

Given the short-term nature of the follow-up, can the authors comment on the potential for recurrence or long-term differences in efficacy among the treatments, especially considering known transient effects of corticosteroids?

Is it possible that the lack of an exercise component in the treatment protocol contributed to the absence of grip strength improvements, and how might the inclusion of strengthening exercises have influenced the outcomes?

Since all three interventions led to significant improvements without clear superiority, what implications do the authors believe this has for cost-effectiveness or patient-centered treatment choices in clinical practice?

While the authors discuss ultrasonographic tendon thickness changes, could they provide more insight into whether the observed short-term reductions are predictive of sustained functional recovery or are merely reflective of transient physiological changes?

Reviewer #2: The sample size calculation section requires more detail. What was the assumed effect size — was it based on a within-group or between-group difference? Was a repeated measures model used? Additionally, was the final sample size adjusted for attrition?

The primary aim is to compare effects between groups. However, the non-significant results are inconclusive, as it is unclear whether they reflect the truth or lack of statistical power.

Line 222: “General Linear Model Repeated Measures” is not a commonly used term. It is recommended to use more standard terminology such as linear mixed model, mixed model for repeated measures (MMRM), or generalized estimating equations (GEE).

Line 218: The reference to "ANOVA (one-way and repetitive)" can be removed, as mixed models offer more flexibility and are generally superior for repeated measures data.

Line 225: Was “adjusted analyses” needs clarification referring to adjustment for covariates? This sentence may be omitted if the analyses were not performed.

Tables: clarify what the “P value **” represents. Is it comparing changes from baseline between groups, or comparisons between groups at each time point? This p value shows the comparisons for baseline also.

**Do you want your identity to be public for this peer review?** For information about this choice, including consent withdrawal, please see our Privacy Policy

Reviewer #1: No

Reviewer #2: No

---

## [Author Response · Author response to Decision Letter 1]

10 Jun 2025

Rebuttal Letter

Title: Response to Reviewers – Manuscript PONE-D-25-13543

Dear Editor,

Thank you for giving us the opportunity to submit a revised draft of our manuscript entitled " Comparison of kinesio tape, counterforce brace, and corticosteroid injection in patients with tennis elbow: a prospective, randomized, controlled study" for publication in the valuable PLOS ONE journal. We appreciate the consideration and effort that you and the reviewers have dedicated to providing your valuable feedback on this manuscript. All authors are grateful to the reviewers for their insightful comments on this paper. Below, we provide a detailed point-by-point response to all reviewer comments. All changes in the revised manuscript have been highlighted using the "Track Changes" feature. We hope that the corrections are satisfied.

Sincerely yours,

Mohammad Taghipour

Corresponding author

Comments from editor:

Comment 1. Make sure the introduction articulates with the research question.

Response: Thank you for this valuable comment. We have revised the Introduction section to clearly articulate the research question and its connection to the rationale of the study.

Location of changes (page/line): Page 4, lines 97-98

Comment 2. Justify the rationale for selecting the reason for selecting these modalities, particularly when the literature shows uncertainty about their mechanism of action and effectiveness. Also there should be a strong need for the study to be conducted as there is adequate literature demonstrating clinical significance and novelty of the study.

Response: Thank you for raising this important point. In the revised manuscript, we have expanded our rationale for selecting the three treatment modalities—kinesiotape, counterforce brace, and corticosteroid injection—especially given the ongoing uncertainty surrounding their mechanisms of action. We also highlighted the novelty and clinical relevance of our study by addressing a key gap in the literature: the lack of a single randomized trial comparing all three interventions. Additionally, the incorporation of ultrasonographic assessment of tendon thickness as an objective and reproducible measure enhances the methodological rigor and adds structural validity to our outcomes. This distinguishes our study from previous research that has relied primarily on subjective assessments.

Location of changes (page/line): (Page 3, lines 72-73) and (Page 4, lines 89-93)

Comment 3. Please make sure the potential confounding variables are addressed in the manuscript.

Response: Thank you for your inquiry. In our study, we used randomization and limitations to control the confounding effect and checked the differences in baseline. We considered three factors(age, gender, BMI) as the most important confounding factors, checked them, and reported them in the results section.

Comment 4. Please justify the sample size calculation. In terms of effect size assumed, which model was used and was patient drop out considered during sample size calculation.

Response: Thank you for your constructive feedback. In this study, the sample size was obtained using G*POWER software. According to the article result (Oken et al), the effect size of pain intensity was considered to be 0.15. Assuming an error of 0.05, a power of 80% for three assessments, and an intra-individual correlation of 0.6, the minimum number of samples required in each group was 15 people. Including a 10% attrition rate, 51 people will be included in the study (17 people in each group)

Thank you again for your valuable feedback, which has contributed to enhancing the rigor and comprehensiveness of our study.

Location of changes (page/line): Page 5, lines 112-117

Comment 5. As there was no exercise protocol included in the intervention period, please justify how its impact on the findings and its application in the real world.

Response: Thank you for your comment and interest in our study. In the revised manuscript, we have explained that the omission of an exercise program was a deliberate decision to isolate the effects of each intervention (kinesio tape, counterforce brace, or corticosteroid injection) without the influence of exercise-based rehabilitation. We also noted that in real-world clinical settings, some patients may be unable or unwilling to participate in exercise programs due to time constraints or physical limitations. Therefore, our findings reflect the standalone effectiveness of each intervention in such scenarios. Nevertheless, we acknowledge in the Discussion that the absence of strengthening exercises may have contributed to the lack of improvement in grip strength, and we recommend that future studies explore combined treatment strategies.

Location of changes (page/line): Page 17, lines 331-335

Comment 6. Please provide detailed explanation on how the authors made sure patient adherence and protocol compliance across three groups, particularly in KT and brace usage group.

Response: We greatly appreciate your dedicated time and effort to providing this interesting feedback. In the revised manuscript, we have included a description of the measures taken to ensure patient adherence and compliance. At the beginning of the intervention, patients in both the kinesio tape and brace groups received thorough verbal instructions regarding proper usage and care. For the kinesio tape group, this included guidance on tape hygiene and minimizing water exposure. In the brace group, patients were educated on correct placement and daily usage duration. Additionally, the treating clinician maintained regular telephone follow-ups with all participants to reinforce protocol adherence and address any concerns. These combined efforts were effective in maintaining a high level of compliance.

Location of changes (page/line): Page 8, lines 172-175

Comment 7. Please explain the measures taken to mitigate potential placebo effects.

Response: Thank you for bringing up this concern. Although a formal placebo control group was not included, we implemented several measures to minimize potential placebo effects. First, all patients received similar neutral information about the effectiveness of each treatment modality, and no emphasis was placed on the superiority of any intervention. Second, outcome assessments were conducted by a blinded evaluator who was unaware of group assignments, helping to avoid assessor bias. Lastly, since all three interventions involved tangible, physically administered treatments (brace, taping, and injection), the likelihood of differential patient expectations was reduced.

Location of changes (page/line): Page 8, lines 180-182

Comment 8. As the blinding was not done, please explain how patient bias was limited.

Response: Thank you for your beneficial suggestion. We have revised the manuscript to clarify how we addressed the potential for patient bias despite the lack of blinding. All patients received standardized, neutral information about the efficacy of the interventions, emphasizing that all three treatments were considered effective and widely accepted for managing lateral epicondylitis. This approach was intended to minimize differential expectations. Furthermore, outcome assessments were conducted by a blinded evaluator to reduce bias in data collection. Lastly, the tangible and active nature of the interventions (brace, taping, and injection) likely reduced the risk of major discrepancies in patient perceptions across groups.

Location of changes (page/line): Page 8, lines 180-182

Comment 9. Please expand on the significance of the findings, especially about the grip strength.

Response: Thank you for this important point. In the revised manuscript, we have highlighted that despite the improvements in pain and tendon thickness, none of the interventions resulted in significant improvements in grip strength. This finding underscores that pain reduction does not necessarily translate into restored muscular function. Clinically, this suggests that additional interventions, such as strengthening exercises, may be necessary to fully restore hand function. Moreover, it emphasizes that achieving pain relief alone should not be regarded as the sole indicator of successful treatment in lateral epicondylitis management.

Location of changes (page/line): Page 17, lines 331-335

Comment 10. As the follow-up was short term, please explain the long-term implications of the treatment on patient outcomes.

Response: Thank you for highlighting this limitation. In the revised Discussion, we have acknowledged the short duration of follow-up as a limitation of our study. Previous research has indicated that certain interventions, such as corticosteroid injections, may yield rapid improvements that are not sustained over time and could be associated with symptom recurrence. Therefore, while our results provide reliable insights into short-term efficacy, they may not fully capture the long-term outcomes of the interventions. We recommend that future studies incorporate extended follow-up periods such as 3, 6, or 12 months to assess both functional and ultrasonographic outcomes over time.

Location of changes (page/line): (Page 17, lines 344-349) and (Page 17, lines 350-351)

Comment 11. Please explain how lack of exercise components may have influenced the grip strength.

Response: Thank you for this comment. In the revised manuscript, we have discussed that the lack of an exercise component particularly strengthening exercises for the wrist and forearm extensors likely contributed to the absence of improvement in grip strength across groups. While the interventions reduced pain and tendon thickness, they may not have been sufficient to restore muscle function. We have acknowledged this limitation and highlighted it as an area for future research.

Location of changes (page/line): Page 17, lines 331-335

Comment 12. Please explain the clinical implication of the results, as all the three interventions had similar effects.

Response: Thank you for this important question. In the revised Discussion, we have elaborated that the similar short-term efficacy of the three interventions suggests that treatment choices for lateral epicondylitis can be tailored to patient preferences, cost considerations, availability, and individual circumstances. For example, patients who prefer to avoid injections may opt for non-invasive treatments like a brace or kinesio tape. Likewise, the counterforce brace, being a low-cost and easily accessible intervention, could be particularly valuable in resource-limited settings. Ultimately, our findings emphasize a patient-centered approach to treatment selection, acknowledging that no single intervention demonstrated clear superiority.

Location of changes (page/line): Page 18, lines 356-357

Comment 13. Please explain if the tendon thickness changes have long term implications on functional recovery or it is a transient change.

Response: Thank you for raising this point. In the revised manuscript, we have clarified that the observed reductions in tendon thickness are likely short-term changes, primarily reflecting reductions in inflammation and local edema. Existing evidence suggests that such short-term ultrasonographic changes may not directly translate into sustained functional recovery. Therefore, longer-term studies are necessary to determine whether these tendon thickness changes correlate with lasting clinical improvements. We have addressed this limitation and the need for further research in the Discussion section.

Location of changes (page/line): Page 16, lines 326-327

Comment 14. Please provide a detailed explanation on the statistically non-significant results.

Response: Thank you for this important comment. In the revised manuscript, we have addressed the non-significant results by explaining that the lack of statistically significant differences, especially in between-group comparisons, may be attributed to factors such as small effect sizes, limited sample size, and the short follow-up duration. These findings do not necessarily imply that no true differences exist between interventions; rather, they may reflect study design limitations. Therefore, we have emphasized that our results should be interpreted with caution and that future studies with larger sample sizes and longer follow-up periods are needed to confirm these findings.

Location of changes (page/line): Page 17, lines 344-349

Reviewers' comments:

Reviewer #1

Comment 1. Does the introduction clearly articulate a specific hypothesis or research question that guides the comparative analysis of the three interventions (KT, counterforce brace, and CSI)?

Response: Thank you for this comment. We have revised the final paragraph of the Introduction to clearly articulate the research question, which focuses on comparing the short-term effects of kinesiotape, counterforce bracing, and corticosteroid injection on pain, tendon thickness, grip strength, and functional status in patients with lateral epicondylitis. Additionally, we have emphasized that the novelty of this study lies not only in the simultaneous comparison of three widely used interventions, but also in the incorporation of ultrasonographic assessment as an objective method to quantify tendon changes. This approach enhances the methodological rigor and clinical relevance of our findings, distinguishing our work from previous studies that relied primarily on subjective outcome measures.

Location of changes (page/line): (Page 4, lines 89-93) and (page4, lines 97-98)

Comment 2. Have the authors sufficiently justified the rationale for selecting these three particular treatment modalities, especially given the noted uncertainty in their mechanisms and long-term effectiveness?

Response: Thank you for the insightful comment. In the revised Introduction, we have sufficiently justified the rationale for selecting these three treatment modalities. Kinesiotape, counterforce brace, and corticosteroid injection are among the most commonly used interventions for lateral epicondylitis in clinical practice. While there is ongoing debate about their precise mechanisms of action and long-term effectiveness, our study was designed to evaluate and compare their short-term effects. This approach is clinically relevant, especially in scenarios where quick symptom relief is desired. We have acknowledged in the Discussion that longer-term outcomes remain uncertain and should be addressed in future research

Location of changes (page/line): (Page 3, lines 72-73) and (page4, lines 89-93)

Comment 3. While prior literature is cited regarding the benefits and limitations of each treatment, do the authors adequately address potential confounding variables or population differences that may have influenced past findings?

Response: Thank you for your inquiry. In our study, we used randomization and limitations to control the confounding effect and checked the differences in baseline. We considered three factors(age, gender, BMI) as the most important confounding factors, checked them, and reported them in the results section.

Comment 4. Given the mention that no prior study has compared all three interventions simultaneously, could the authors elaborate more on the clinical significance and novelty of this comparison to reinforce the study’s contribution to the field?

Response: Thank you for this important point. In the revised manuscript, we have emphasized both the novelty and clinical significance of our study. To the best of our knowledge, no previous randomized controlled trial has simultaneously compared all three interventions, highlighting the contribution of our findings to evidence-based clinical decision-making. Moreover, we have clarified that the study’s innovation extends beyond the comparative design to include the use of ultrasonographic assessment as an objective tool for quantifying tendon changes. This methodological addition enhances the scientific robustness of the study and offers structural insights that go beyond subjective outcome measures used in earlier research.

Location of changes (page/line): Page 4, (lines 89-93)

Comment 5. Is the justification for the sample size calculation sufficiently clear, particularly

---

## [Decision Letter · Decision Letter 1]

Comparison of kinesio tape, counterforce brace, and corticosteroid injection in patients with tennis elbow: a prospective, randomized, controlled study

PONE-D-25-13543R1

Dear Dr. Taghipour,

We’re pleased to inform you that your manuscript has been judged scientifically suitable for publication and will be formally accepted for publication once it meets all outstanding technical requirements.

Kind regards,

Prateek Srivastav

Academic Editor

PLOS ONE

Additional Editor Comments (optional):

Reviewers' comments:

Reviewer's Responses to Questions

**Comments to the Author**

Reviewer #1: All comments have been addressed

Reviewer #2: All comments have been addressed

2. Is the manuscript technically sound, and do the data support the conclusions?

Reviewer #1: Yes

Reviewer #2: (No Response)

3. Has the statistical analysis been performed appropriately and rigorously?

Reviewer #1: Yes

Reviewer #2: (No Response)

4. Have the authors made all data underlying the findings in their manuscript fully available?

Reviewer #1: Yes

Reviewer #2: (No Response)

5. Is the manuscript presented in an intelligible fashion and written in standard English?

Reviewer #1: Yes

Reviewer #2: (No Response)

Reviewer #1: Dear Author,

Your article titled "Comparison of kinesiotape, counterforce brace, and corticosteroid injection in patients with tennis elbow" presents a valuable comparison of three widely used interventions in the treatment of lateral epicondylitis. The prospective and randomized nature of the study is a key advantage. I have reviewed the revised version and I agree with the changes you have made. I support the acceptance and publication of your manuscript.

Reviewer #2: All my concerns are addressed.

**Do you want your identity to be public for this peer review?** For information about this choice, including consent withdrawal, please see our Privacy Policy

Reviewer #1: No

Reviewer #2: No

---

## [Editor Report · Acceptance letter]

PONE-D-25-13543R1

PLOS ONE

Dear Dr. Taghipour,

I'm pleased to inform you that your manuscript has been deemed suitable for publication in PLOS ONE. Congratulations! Your manuscript is now being handed over to our production team.

Kind regards,

on behalf of

Dr. Prateek Srivastav

Academic Editor

PLOS ONE